# Policy Gradient With Serial Markov Chain Reasoning

**Edoardo Cetin**
Department of Engineering
King's College London
edoardo.cetin@kcl.ac.uk

**Oya Celiktutan**
Department of Engineering
King's College London
oya.celiktutan@kcl.ac.uk

## Abstract

We introduce a new framework that performs decision-making in reinforcement learning (RL) as an iterative *reasoning* process. We model agent behavior as the *steady-state distribution* of a parameterized *reasoning* Markov chain (RMC), optimized with a new tractable estimate of the policy gradient. We perform action selection by simulating the RMC for enough *reasoning steps* to approach its steady-state distribution. We show our framework has several useful properties that are inherently missing from traditional RL. For instance, it allows agent behavior to approximate any continuous distribution over actions by parameterizing the RMC with a simple Gaussian transition function. Moreover, the number of reasoning steps to reach convergence can scale adaptively with the difficulty of each action selection decision and can be accelerated by re-using past solutions. Our resulting algorithm achieves state-of-the-art performance in popular Mujoco and DeepMind Control benchmarks, both for proprioceptive and pixel-based tasks.

## 1 Introduction

Reinforcement learning (RL) has the potential to provide a general and effective solution to many modern challenges. Recently, this class of methods achieved numerous impressive milestones in different problem domains, such as games [1–3], robotics [4–6], and other meaningful real-world applications [7–9]. However, all these achievements relied on massive amounts of data, controlled environments, and domain-specific tuning. These commonalities highlight some of the current practical limitations that prevent RL to be widely applicable [10].

In the deep RL framework, practitioners train agents with the end goal of obtaining optimal *behavior*. Traditionally, agent behavior is modeled with *feed-forward policies* regressing from any state to a corresponding distribution over actions. Such formulation yields practical training objectives in both off-policy [11–13] and on-policy settings [14–16]. However, we identify three inherent properties of this rigid representation of behavior that could considerably impact expressivity and efficiency in continuous control tasks. First, agent behavior is restricted to a class of tractable distributions, which might fail to capture the necessary complexity and multi-modality of a task. Second, the policy performs a fixed *reasoning* process with a feed-forward computation, which potency *cannot adapt* to the varying complexity of individual action selection problems. Third, decision-making is performed every time *from scratch*, without re-using any past information that might still inform and facilitate the current action selection problem.

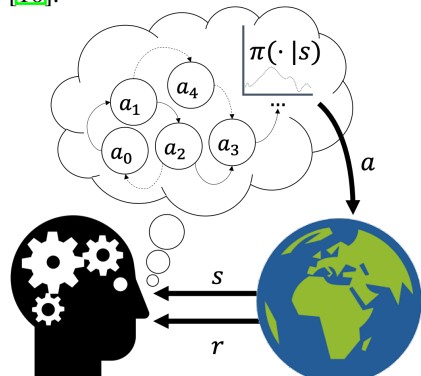

Figure 1: Depiction of agent decision-making with serial Markov chain reasoning.

Unlike RL policies, human reasoning does not appear to follow a rigid feed-forward structure. In fact, a range of popular psychological models characterize human decision-making as a *sequential* process with adaptive temporal dynamics [17–20]. Many of these models have found empirical groundings in neuroscience [21–24] and have shown to effectively complement RL for capturing human behavior in experimental settings [25, 26]. Partly inspired by these works, we attempt to *reframe* the deep RL framework by making use of a similar flexible model of agent behavior, in order to counteract its aforementioned limitations.

We introduce *serial Markov chain reasoning* - a new powerful framework for representing agent behavior. Our framework treats decision-making as an *adaptive reasoning process*, where the agent sequentially updates its beliefs regarding which action to execute in a series of *reasoning steps*. We model this process by replacing the traditional policy with a parameterized *transition function*, which defines a *reasoning Markov chain* (RMC). The steady-state distribution of the RMC represents the distribution of agent behavior after performing enough *reasoning* for decision-making. Our framework naturally overcomes the aforementioned limitations of traditional RL. In particular, we show that our agent's behavior can approximate any arbitrary distribution even with simple parameterized transition functions. Moreover, the required number of *reasoning steps* adaptively scales with the difficulty of individual action selection problems and can be accelerated by re-using samples from similar RMCs.

To optimize behavior modeled by the steady-state distribution of the RMC, we derive a new tractable method to estimate the *policy gradient*. Hence, we implement a new effective off-policy algorithm for maximum entropy reinforcement learning (MaxEnt RL) [27, 28], named *Steady-State Policy Gradient* (SSPG). Using SSPG, we empirically validate the conceptual properties of our framework over traditional MaxEnt RL. Moreover, we obtain state-of-the-art results for popular benchmarks from the OpenAI Gym Mujoco suite [29] and the DeepMind Control suite from pixels [30].

In summary, this work makes the following key contributions:

1. We propose *serial Markov Chain reasoning* a framework to represent agent behavior that can overcome expressivity and efficiency limitations inherent to traditional reinforcement learning.

2. Based on our framework, we derive *SSPG*, a new tractable off-policy algorithm for MaxEnt RL.

3. We provide experimental results validating theorized properties of *serial Markov Chain reasoning* and displaying state-of-the-art performance on the Mujoco and DeepMind Control suites.

## 2 Background

### 2.1 Reinforcement learning problem

We consider the classical formulation of the reinforcement learning (RL) problem setting as a Markov Decision Process (MDP) [31], defined by the tuple $(S, A, P, p_0, r, \gamma)$. In particular, at each discrete time step $t$ the agent experiences a state from the environment's state-space, $s_t \in S$, based on which it selects an action from its own action space, $a_t \in A$. In continuous control problems (considered in this work), the action space is typically a compact subset of an Euclidean space $\mathbb{R}^{dim(A)}$. The evolution of the environment's state through time is determined by the transition dynamics and initial state distribution, $P$ and $p_0$. Lastly, the reward function $r$ represents the immediate level of progress for any state-action tuple towards solving a target task. The agent's behavior is represented by a state-conditioned parameterized policy distribution $\pi_\theta$. Hence, its interaction with the environment produces trajectories, $\tau = (s_0, a_0, s_1, ..., s_T, a_T)$, according to a factored joint distribution $p_{\pi_\theta}(\tau) = p_0(s_0) \prod_{t=0}^{T} \pi_\theta(a_t|s_t)P(s_{t+1}|s_t, a_t)$. The RL objective is to optimize agent behavior as to maximize the discounted sum of expected future rewards: $\arg\max_\theta \mathbb{E}_{p_{\pi_\theta}(\tau)} \left[ \sum_{t=0}^{T} \gamma^t r(s_t, a_t) \right]$.

### 2.2 Maximum entropy reinforcement learning and inference

Maximum entropy reinforcement learning (MaxEnt RL) [32] considers optimizing agent behavior for a different objective that naturally arises when formulating action selection as an inference problem [33–36]. Following Levine [28], we consider modeling a set of binary optimality random variables with realization probability proportional to the exponentiated rewards scaled by the temperature $\alpha$, $p(O_t|s_t, a_t) \propto \exp(\frac{1}{\alpha}r(s_t, a_t))$. The goal of MaxEnt RL is to minimize the KL-divergence between

trajectories stemming from agent behavior, $p_{\pi_\theta}(\tau)$, and the inferred optimal behavior, $p(\tau|O_{0:T})$:

$$
\begin{aligned}
D_{KL}\left(p_{\pi_\theta}(\tau)||p(\tau|O_{0:T})\right) &= \mathbb{E}_{p_{\pi_\theta}(\tau)}\left[\log\frac{p_0(s_0)\prod_{t=0}^{T}\pi_\theta(a_t|s_t)P(s_{t+1}|s_t,a_t)}{p_0(s_0)\prod_{t=0}^{T}\exp(\frac{1}{\alpha}r(s_t,a_t))P(s_{t+1}|s_t,a_t)}\right] \\
&= -\mathbb{E}_{p_{\pi_\theta}(\tau)}\left[\sum_{t=0}^{T}r(s_t,a_t)+\alpha H(\pi(\cdot|s_t))\right].
\end{aligned}
\tag{1}
$$

The resulting entropy-regularized objective introduces an explicit trade-off between exploitation and exploration, regulated by the temperature parameter $\alpha$ scaling the policy's entropy. An effective choice to optimize this objective is to learn an auxiliary parameterized *soft Q-function* [37]:

$$
Q_\phi^\pi(s_t,a_t) = \mathbb{E}_{p_{\pi_\theta}(\tau|s_t,a_t)}\left[r(s_t,a_t)+\sum_{t'=t+1}^{T}r(s_{t'},a_{t'})+\alpha H(\pi(a_{t'}|s_{t'}))\right].
\tag{2}
$$

Given some state, $Q_\phi^\pi(s,\cdot)$ represents an energy-function based on the expected immediate reward and the agent's future likelihood of optimality from performing any action. Thus, we can locally optimize the MaxEnt objective by reducing the KL-divergence between $\pi$ and the canonical distribution of its current soft Q-function. This is equivalent to maximizing the expected soft Q-function's value corrected by the policy's entropy, resembling a regularized policy gradient objective [11, 12]:

$$
\arg\max_\theta \mathbb{E}_{s,a\sim\pi_\theta(\cdot|s)}\left[Q_\phi^\pi(s,a)+\alpha H(\pi_\theta(a|s))\right].
\tag{3}
$$

The policy is usually modeled with a neural network outputting the parameters of some tractable distribution, such as a factorized Gaussian, $\pi_\theta(\cdot|s) = N(\mu_\theta(s);\Sigma_\theta(s))$. This practice allows to efficiently approximate the gradients from Eqn. 3 via the reparameterization trick [38]. We consider the off-policy RL setting, where the agent alternates learning with storing new experience in a data buffer, $D$. We refer the reader to Haarnoja et al. [13, 39] for further derivation and practical details.

## 3 Policy Gradient with serial reasoning

### 3.1 Reasoning as a Markov chain

We introduce *Serial Markov Chain Reasoning*, a new framework to model agent behavior, based on conceptualizing action selection as an adaptive, sequential process which we refer to as *reasoning*. Instead of using a traditional policy, the agent selects which action to execute by maintaining an internal *action-belief* and a *belief transition (BT-) policy*, $\pi^b(a'|a,s)$. During the *reasoning process*, the agent updates its action-belief for a series of *reasoning steps* by sampling a new action with the BT-policy $\pi^b$ taking both environment state and previous action-belief as input. We naturally represent this process with a *reasoning Markov chain* (RMC), a discrete-time Markov chain over different action-beliefs, with transition dynamics given by the BT-policy. Hence, for any input environment state $s$ and initial action-belief $a_0$, the n-step transition probabilities of the RMC for future *reasoning steps* $n = 1, 2, 3, ...$ are defined as:

$$
\pi_n^b(a|a_0,s) = \int_A \pi^b(a|a',s)\pi_{n-1}^b(a'|a_0,s)da', \quad \text{for } n>1, \text{ and } \quad \pi_1^b = \pi^b.
\tag{4}
$$

Given a compact action space and a BT-policy with a non-zero infimum density, we can ensure that as the number of reasoning steps grows, the probability of any action-belief in the RMC *converges* to some *steady-state probability* which is independent of the initial action-belief.[1] We denote this *implicit* probability distribution as the *steady-state (SS-) policy*, symbolized by $\pi^s(a|s)$:

**Lemma 3.1.** ***Steady-state convergence.*** *For any environment state $s$, consider a reasoning Markov chain (RMC) defined on a compact action space $A$ with transition probabilities given by $\pi^b(a'|a,s)$. Suppose that $\inf\{\pi^b(a'|a,s) : a', a \in A\} > 0$. Then there exists a steady-state probability distribution function $\pi^s(\cdot|s)$ such that:*

$$
\lim_{n\to\infty}\pi_n^b(a|a_0,s) \to \pi^s(a|s) \quad \text{for all } a \in A.
\tag{5}
$$

*Proof.* See Appendix A. $\square$

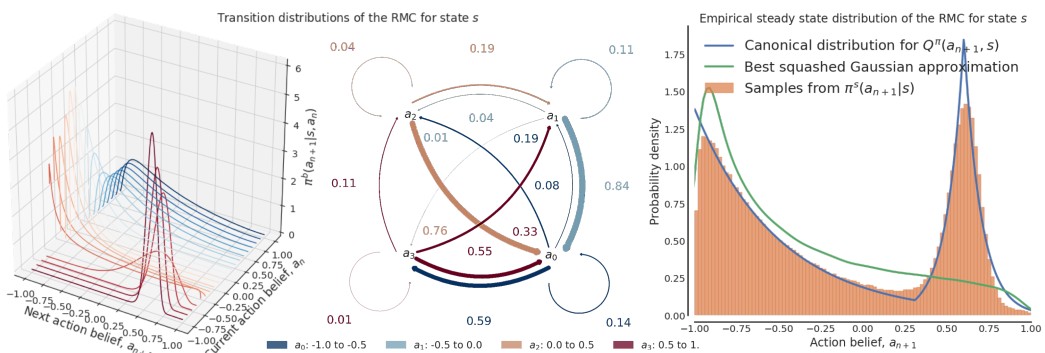

Figure 2: (**Left**) BT-policy transition probabilities and quantized RMC state diagram. (**Right**) Sample approximation of related steady-state distribution as compared to canonical distribution of the soft Q-function.

The RMC's steady-state probabilities can be interpreted as representing the distribution of agent's behavior after an *appropriate* number of reasoning steps are performed. In this work, we strive to optimize the agent's behavior following the MaxEnt RL framework described in Section 2. In particular, we consider learning a parameterized BT-policy, $\pi_\theta^b$, to produce appropriate transition probabilities for each environment state such that the SS-policy, $\pi_\theta^s$, from the resulting RMC optimizes:

$$\arg\max_\theta J(\theta) = \mathbb{E}_{s,a\sim\pi_\theta^s(\cdot|s)}\left[Q_\phi^s(s,a) + \alpha H(\pi_\theta^s(a|s))\right]. \tag{6}$$

Here, $Q_\phi^s$ is a parameterized soft Q-function for the agent's behavior from $\pi^s$, which we learn by minimizing a squared *soft* Bellman loss utilizing delayed parameters $\phi'$ and samples from $\pi_\theta^s$:

$$\arg\min_\phi J(\phi) = \mathbb{E}_{s,a,s'}\left[\left(Q_\phi^s(s,a) - \left(r(s,a) + \gamma\,\mathbb{E}_{a'\sim\pi_\theta^s(\cdot|s)}\left[Q_{\phi'}^s(s',a') + \alpha H(\pi_\theta^s(a'|s'))\right]\right)\right)^2\right]. \tag{7}$$

In Fig. 2, we illustrate the relationship between a learned BT-policy, the corresponding SS-policy, and the soft Q-function in a 1-dimensional toy task (see App. C for details). In this example, the BT-policy is parameterized as a simple squashed Gaussian distribution, with unimodal transitions between consecutive action beliefs (Fig. 2, Left). We obtain samples of agent behavior (the SS-policy) by performing a series of reasoning steps, using the BT-policy to simulate the RMC until we approach steady-state convergence. By plotting the resulting empirical distribution of agent behavior, we see it closely matches the multi-modal, non-Gaussian canonical distribution from its soft Q-function (Fig. 2, Right). This example shows how the expressive power of agent behavior in our framework can go far beyond the BT-policy's simple parameterization, enabling for the effective maximization of complex and multi-modal MaxEnt objectives.

## 3.2 Learning the belief transition policy

We propose a new method to estimate the policy gradient of the BT-policy, $\pi_\theta^b$, for optimizing the steady-state MaxEnt objective described in Section 3.1. We note that the gradient from Eq. 6 involves differentiating through an expectation of the steady-state policy, $\pi_\theta^s$. However, $\pi_\theta^s$ is only *implicitly* defined, and its connection with the actual BT-policy or its parameters does not have a tractable closed-form expression. To approach this problem, we introduce a family of *n-step extensions* to the soft Q-function, $Q_n^s : S \times A \mapsto \mathbb{R}$ for $n = 0, 1, 2, \ldots$, defined as:

$$Q_n^s(s,a) = \int_A \pi_n^b(a'|a,s)Q_\phi^s(s,a')da', \quad with \quad \nabla_\theta Q_n^s(s,a) = \mathbf{0}. \tag{8}$$

Intuitively, each *n-step soft Q-function* $Q_n^s(s,a)$ outputs the *expected soft Q-value* after performing $n$ reasoning steps in the RMC from the initial action-belief $a$. However, we treat the output of each n-step soft Q-function as being *independent* of the actual parameters of the BT-policy, $\theta$. Hence, we can interpret computing $Q_n^s(s,a)$ as simulating the RMC with a *fixed and immutable copy* of the current $\pi_\theta^b$. We use this definition to provide a convenient notation in the following new Theorem that expresses the policy gradient without differentiating through $\pi_\theta^s$:

---

[1]This is unrelated to the *steady-state* distribution for infinite-horizon MDPs considered in prior work [40].

**Theorem 3.2.** *Steady-state policy gradient.* *Let $\pi_\theta^b(\cdot|a, s)$ be a parameterized belief transition policy which defines a reasoning Markov chain with a stationary distribution given by the steady-state policy $\pi_\theta^s(\cdot|s)$. Let $Q^s$ be a real function defined on $S \times A$, with a family of n-step extensions $\{Q_n^s\}$ as defined in Eq. 8. Suppose $\pi^b$, $Q^s$ and their gradient with respect to $\theta$ (denoted $\nabla_\theta$) are continuous and bounded functions. Then*

$$\nabla_\theta \mathbb{E}_{a \sim \pi_\theta^s(\cdot|s)} [Q^s(s, a)] = \mathbb{E}_{a \sim \pi_\theta^s(\cdot|s)} \left[ \lim_{N \to \infty} \sum_{n=0}^{N} \nabla_\theta \mathbb{E}_{a' \sim \pi_\theta^b(\cdot|a,s)} [Q_n^s(s, a')] \right]. \qquad (9)$$

*Proof.* See Appendix A. $\qquad\qquad\qquad\qquad\qquad\qquad\qquad\qquad\qquad\qquad\qquad\qquad\qquad\qquad$ □

Using Lemma 3.1 (*steady-state convergence*), we can approximate the policy gradient expression in Eq. 9 with an arbitrarily small expected error using a finite number of n-step soft Q-functions, i.e., $N$ (see App. A). An intuition for this property follows from the fact that for large enough $n$, Lemma 3.1 implies that $\pi_n^b(a|a', s) \approx \pi_\theta^s(a|s)$ and, thus, $Q_n^s(s, a') \approx \int_A \pi_\theta^b s(a|s) Q_\phi^s(s, a) da$. Therefore, the value of each $Q_n^s(s, a')$ will be independent of the BT-policy's action $a'$, such that $\nabla_\theta \mathbb{E}_{a' \sim \pi_\theta^b(\cdot|a,s)} [Q_n^s(s, a')] \approx \mathbf{0}$. In other words, each subsequent step in the RMC introduces additional randomness that is independent of $a'$, causing a warranted *vanishing gradient* phenomenon [41] which culminates with converging to $\pi_\theta^s$. Using a similar notation as Haarnoja et al. [39], we apply the reparameterization trick [38] to express the BT-policy in terms of a deterministic function $f_\theta^b(a, s, \epsilon)$, taking as input a Gaussian noise vector $\epsilon$. This allows to rewrite the gradient in each inner expectation in the sum from Eq. 9 as:

$$\nabla_\theta \mathbb{E}_{a' \sim \pi_\theta^b(\cdot|a,s)} [Q_n^s(s, a')] = \mathbb{E}_{\epsilon_0 \sim N(0,1)} \left[ \nabla_{a_0} Q_n^s(s, a_0) \nabla_\theta f_\theta^b(a, s, \epsilon_0) \right], \qquad (10)$$

where $a_0 = f_\theta^b(a, s, \epsilon_0)$. We can apply the same reparameterization for all n-step soft Q-functions, to establish a new relationship between the gradient terms $\nabla_{a_0} Q_n^s(s, a_0)$:

$$\nabla_{a_0} Q_n^s(s, a_0) = \nabla_{a_0} \int_A \pi_n^b(a_n|a_0, s) Q_\phi^s(s, a_n) da_n = \nabla_{a_0} \int_A \pi^b(a_1|a_0, s) Q_{n-1}^s(s, a_1) da_1$$
$$= \mathbb{E}_{\epsilon_1} \left[ \nabla_{a_1} Q_{n-1}^s(s, a_1) \nabla_{a_0} f^b(a_0, s, \epsilon_1) \right], \quad where, \ a_1 = f(a_0, s, \epsilon_1). \qquad (11)$$

In Eq. 11, we purposefully omit the dependence of $f^b$ and $\pi^b$ from $\theta$ since each $Q_n^s$ term is a *local approximation* of the RMC that does not depend on $\theta$ (as defined in Eq. 8). By recursively applying this relationship (Eq. 11) to $\nabla_{a_1} Q_{n-1}^s(s, a_1)$ and all subsequent gradient terms we obtain:

$$\nabla_{a_0} Q_n^s(s, a_0) = \mathbb{E}_{\epsilon_1, \dots, \epsilon_n} \left[ \nabla_{a_n} Q_\phi^s(s, a_n) \prod_{i=0}^{n-1} \nabla_{a_i} f^b(a_i, s, \epsilon_{i+1}) \right], \qquad (12)$$

where $a_i = f^b(a_{i-1}, s, \epsilon_i)$ for $i = 1, \dots, n$. By combining Eq. 10 and Eq. 12, we can thus reparameterize and express the whole sum in Eq. 9 as:

$$\nabla_\theta \mathbb{E}_{a \sim \pi_\theta^s(\cdot|s)} [Q^s(s, a)] \approx \sum_{n=0}^{N} \nabla_\theta \mathbb{E}_{a' \sim \pi_\theta^b(\cdot|a,s)} [Q_n^s(s, a')]$$
$$= \mathbb{E}_{\epsilon_0, \dots, \epsilon_N} \left[ \sum_{n=0}^{N} \nabla_{a_n} Q_\phi^s(s, a_n) \left( \prod_{i=0}^{n-1} \nabla_{a_i} f^b(a_i, s, \epsilon_{i+1}) \right) \nabla_\theta f_\theta^b(a, s, \epsilon_0) \right]. \qquad (13)$$

Eq. 13 intuitively corresponds to differentiating through each $Q_n^s(s, a')$ term by *reparameterizing the RMC*. Hence, to get a sample estimate of the policy gradient we can simulate the reparameterized RMC for $N$ reasoning steps to obtain $a_1, \dots, a_N$, compute each $Q_\phi^s(s, a_n)$ term, and backpropagate (e.g., with *autodifferentiation*). Following Haarnoja et al. [13, 39], we can apply Theorem 3.2 and easily extend the same methodology to estimate the MaxEnt policy gradient from Eq. 6 that also involves an extra entropy term. We include this alternative derivation in App. A for completeness.

| **Algorithm 1** Agent Acting | **Algorithm 2** Agent Learning |
|---|---|

**Algorithm 1** Agent Acting

**input:** $s$, current state
$\mathbf{a_0} \sim \hat{A}$
$N \leftarrow 0$
$R^p \leftarrow +\infty$
**while** $R^p > 1.1$ **do**
$\quad \mathbf{a_{N+1}} \sim \pi_\theta^b(\cdot | \mathbf{a_N})$
$\quad N \leftarrow N + 1$
$\quad$ Update $R^p$ with $\mathbf{a_{1:N}}$ $\qquad \triangleright$ Eq. 16
$\hat{N} \leftarrow \rho \hat{N} + (1 - \rho) N$ $\qquad \triangleright \rho \in [0, 1)$
$\hat{A} \leftarrow \hat{A} \cup \mathbf{a_{1:N}}$
**output:** $a \sim \mathbf{a_{1:N}}$

**Algorithm 2** Agent Learning

**input:** $D$, data buffer
$(s, a, s', r) \sim D$
$a_0 \sim \pi_\theta^b(\cdot | a, s')$
**for** $n \leftarrow 0, \lceil \hat{N} \rceil$ **do**
$\quad Q_n^s \leftarrow Q_\phi^s(s', a_n)$ $\qquad \triangleright$ Eq. 8
$\quad \epsilon_{n+1} \sim N(0, 1), \quad a_{n+1} = f^b(a_n, s, \epsilon_{n+1})$
$\nabla_\theta Q_\phi^s \leftarrow \nabla_\theta(\sum_{n=0}^{\lceil N \rceil} Q_n^s)$ $\qquad \triangleright$ Thm. 3.2
$\arg\min_\theta J(\theta)$ $\qquad \triangleright$ Eq. 6
$a' \sim a_{1:\lceil \hat{N} \rceil}$
$\arg\min_\phi J(\phi)$ $\qquad \triangleright$ Eq. 7

## 3.3 Action selection and temporal consistency

To collect experience in the environment, we propose to perform reasoning with the BT-policy starting from *a set* of different initial action-beliefs $\{a_0^0, ..., a_0^M\}$. We *batch* this set as a single input matrix, $\mathbf{a_0}$, to make effective use of parallel computation. To reduce the required number of reasoning steps and facilitate detecting convergence to $\pi_\theta^s$, we identify two desirable properties for the distribution of action-beliefs in $\mathbf{a_0}$. In particular, initial action-beliefs should 1) be likely under $\pi_\theta^s$, and 2) cover diverse modes of $\pi_\theta^s$. Property (1) should logically accelerate reasoning by providing the BT-policy with already-useful information about optimal behavior. Property (2) serves to provide the BT-policy with initial information of diverse behavior, which facilitates convergence detection (Sec. 3.4) and expedites reasoning even if the RMC has slow mixing times between multiple modes. To satisfy these properties, we use a simple effective heuristic based on common temporal-consistency properties of MDPs [42, 43]. Especially in continuous environments, actions tend to have small individual effects, making them likely relevant also for environment states experienced in the near future. Thus, we propose storing past action-beliefs in a fixed sized buffer, called the *short-term action memory*, $\hat{A}$, and use them to construct $\mathbf{a_0}$. We find this strategy allows to effectively regulate the initial action-beliefs quality and diversity through the size of $\hat{A}$, accelerating convergence at negligible cost.

## 3.4 Detecting convergence to the steady-state policy

A key requirement for learning and acting with BT-policies, as described in Sections 3.2 and 3.3, is the ability to determine a *sufficient* number of reasoning steps ($N$) for the action-belief distribution to converge. Given the properties of the RMC, there exist different analytical methods that provide a priori bounds on the rate of convergence [44–46]. However, using any fixed $N$ would be extremely limiting as we expect the BT-policy and the properties of its resulting RMCs to continuously evolve during training. Moreover, different tasks, states, and initial action-beliefs might affect the number of reasoning steps required for convergence due to different levels of complexity for the relative decision-making problems. To account for similar conditions, in the Markov Chain Monte Carlo literature, the predominant approach is to perform a statistical analysis of the properties of the simulated chain, choosing from several established convergence diagnostic tools [47–49]. Hence, we propose to employ a similar *adaptive strategy* by analyzing the history of the simulated RMC to determine the appropriate number of reasoning steps. Since we apply $\pi_\theta^b$ from a diverse set of initial action beliefs (see Section 3.3), we base our convergence-detection strategy on the seminal Gelman-Rubin (GR) diagnostic [50] and its multivariate extension [51]. In particular, the multivariate GR diagnostic computes the *pseudo scale reduction factor* (PSRF), a score representing whether the statistics of a multivariate variable of interest have converged to the steady-state distribution. The intuition behind this diagnostic is to compare two different estimators of the covariance for the unknown steady-state distribution, making use of either the samples *within* each individual chain and *between* all different chains. Thus, as the individual chains approach the true steady-state distribution, the two estimates should expectedly get closer to each other. The PSRF measures this precise similarity based on the largest eigenvalue of their matrix product.

For our use-case, we employ the PSRF to determine the convergence of the set of action-beliefs $\mathbf{a_{1:N}}$, as we perform consecutive reasoning steps with $\pi_\theta^b$. Following [51], we calculate the average sample

covariance of the action-beliefs *within* each of the parallel chains ($W$) computed from a batched set of initial action-beliefs $\mathbf{a_0} = [a_0^1, a_0^2, \ldots a_0^M]$:

$$\bar{a}^m = \frac{1}{N} \sum_{n=1}^{N} a_n^m, \quad W_m = \frac{1}{N-1} \sum_{n=1}^{N} (a - \bar{a}^m)(a - \bar{a}^m)^T, \quad W = \frac{1}{M} \sum_{m=1}^{M} W_m. \quad (14)$$

We compare $W$ with an unbiased estimate of the target covariance, constructed from the sample covariance *between* the different parallel chains ($B$):

$$\bar{a} = \frac{1}{N \times M} \sum_{n=1}^{N} \sum_{n=1}^{M} a_n^m, \quad B = \frac{1}{M-1} \sum_{n=1}^{N} (\bar{a}^m - \bar{a})(\bar{a}^m - \bar{a})^T. \quad (15)$$

The PSRF for $\mathbf{a_{1:N}}$ is then computed from the largest eigenvalue ($\lambda_{max}$) of the product $W^{-1}B$, as:

$$R^p = \sqrt{\frac{N-1}{N} + \lambda_{max}(W^{-1}B)}. \quad (16)$$

Thus, as the individual chains approach the distribution of $\pi_\theta^s$, the PSRF ($R^p$) will approach 1. Following Brooks and Gelman [51], we use $R^p < 1.1$ as an effective criterion for determining the convergence of $\mathbf{a_{1:N}}$. In practice, we also keep a *running mean* of the current number of reasoning steps for convergence, $\hat{N}$. We use $\lceil \hat{N} \rceil$ as the number of reasoning steps to simulate the RMC with $\pi_\theta^b$ when computing gradients from Eqs. 6-7. $\lceil \hat{N} \rceil$ is a *safe* choice to ensure near unbiased optimization since $R^p < 1.1$ is considered a very conservative criterion [52] and we can learn by simulating the RMC from recent actions stored in the data buffer, which are already likely close to optimal. We provide further details regarding our implementation and its rationale in App. B. We provide a simplified summary of our adaptive reasoning process for acting and learning in Algs. 1-2.

### 3.5 Advantages of serial Markov chain reasoning

Based on the above specification, we identify three main conceptual advantages of our serial Markov chain reasoning framework. **1. Unlimited expressiveness.** The distribution of agent behavior given by the SS-policy $\pi_\theta^s$, is a mixture model with potentially infinitely many components. Thus, even a simple Gaussian parameterization of the BT-policy $\pi_\theta^b$ would make $\pi_\theta^s$ a *universal approximator of densities*, providing unlimited expressive power to the agent [53, 54]. **2. Adaptive computation.** The number of reasoning steps performed to reach approximate convergence is determined by the properties of each environment state's RMC. Hence, the agent can flexibly spend different amounts of computation time based on the complexity of each action-selection problem, with potential gains in both precision and efficiency. **3. Information reuse.** By storing past solutions to similar RMCs, we can initialize the reasoning process with initial action-beliefs that are already close to $\pi_\theta^s$. This allows using the temporal-consistency properties of the MDP to exploit traditionally discarded information and accelerate agent reasoning. We provide empirical validation for these properties in Section 4.2.

## 4 Experimentation

### 4.1 Performance evaluation

We evaluate the *serial Markov chain reasoning* framework by comparing its performance with current state-of-the-art baselines based on traditional RL. We consider 6 challenging Mujoco tasks from Gym [29, 56] and 12 tasks pixel-based tasks from the DeepMind Control Suite (DMC) [30]. In both settings, we base our implementation on MaxEnt RL, replacing the traditional policy with a Gaussian BT-policy optimized with the training procedures specified in Sec. 3. Other orthogonal design choices (e.g., network architectures) follow contemporary RL practices, we refer to App. C or the code for full details. We call the resulting algorithm *Steady-State Policy Gradient* (SSPG).

We report the mean performance curves and aggregate metrics using the statistical tools from *Rliable* [55]. In particular, we compare normalized *performance profiles* [57], *interquantile mean* (IQM), and *probability of improvements* over baselines with the Mann-Whitney U statistic [58]. The reported ranges/shaded regions represent 95% *stratified bootstrap confidence intervals* (CIs) [59]. In App. D, we provide per-task results and further statistical analysis. For each experiment, we collect the returns of SSPG over five seeds, by performing 100 evaluation rollouts during the last 5% of steps.

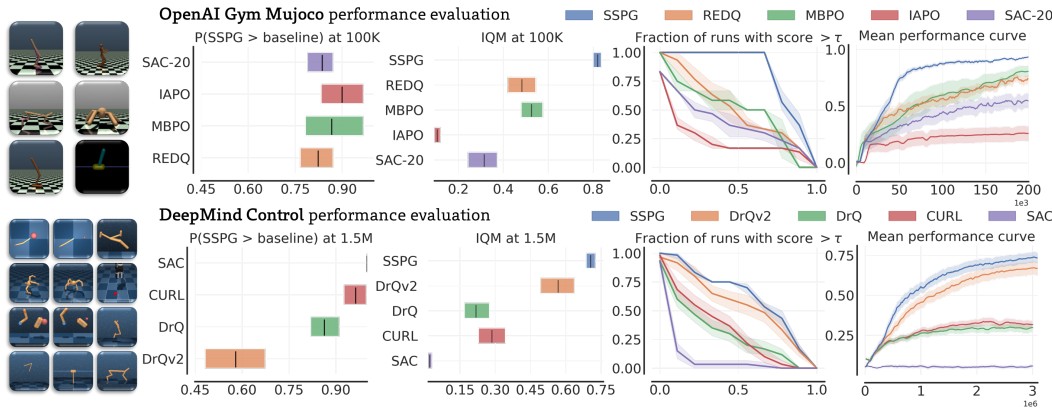

Figure 3: Performance evaluation of SSPG and recent state-of-the-art baselines using *Rliable* [55]. We consider six OpenAI Gym Mujoco tasks [29] (**Top**) and twelve DeepMind Control tasks from pixels [30] (**Bottom**).

**Mujoco suite.** We evaluate on a challenging set of Mujoco tasks popular in recent literature. We compare SSPG with recent RL algorithms achieving state-of-the-art sample-efficiency performance on these tasks, which utilize large critic ensembles and high update-to-data (UTD) ratios. We consider *REDQ* [60] and *MBPO* [61] for state-of-the-art algorithms based on the traditional model-free and model-based RL frameworks. We also compare with *iterative amortized policy optimization* (IAPO) [62], in which the agent performs iterative amortization to optimize its policy distribution [63]. This procedure for action selection is more computationally involved than our agent's *reasoning process*, as it requires both evaluating the policy and computing gradients at several iterations. Yet, as IAPO is still based on the traditional policy gradient framework, its benefits are solely due to reducing the *amortization gap* with an alternative action inference procedure. To ground different results, we also show the performance of the seminal *Soft Actor-Critic* (SAC) algorithm [39], upon which all considered policy gradient baselines are based on. To account for the additional computational cost of training an agent with serial Markov chain reasoning, we use a UTD ratio that is *half* the other algorithms. On our hardware, this makes SSPG faster than all other modern baselines (see App. D).

Figure 3 (Top) shows the performance results after 100K environment steps. Individual scores are normalized using the performance of SAC after 3M steps, enough to reach convergence in most tasks. SSPG considerably outperforms all prior algorithms with *statistically meaningful* gains, as per the conservative Neyman-Pearson statistical testing criterion [64]. Furthermore, SSPG even *stochastically dominates* all considered state-of-the-art baselines [65]. We obtain similar results evaluating at 50K and 200K steps (App. D). In comparison, IAPO obtains lower performance than other non-iterative baselines while being the most compute-intensive algorithm. This indicates that, for sample-efficiency, only reducing the amortization gap beyond direct estimation might not provide significant benefits. Instead, serial Markov chain reasoning's improved expressivity and flexibility appear to considerably accelerate learning, yielding state-of-the-art performance in complex tasks.

**DeepMind Control suite.** To validate the generality of our framework, we also evaluate on a considerably different set of problems: 12 *pixel-based* DMC tasks. We follow the recent task specifications and evaluation protocols introduced by Yarats et al. [66]. We compare SSPG with *DrQv2* [66], the current state-of-the-art policy gradient algorithm on this benchmark, which employs a deterministic actor and hand-tuned exploration. We also compare with additional baselines that, like SSPG, are based on MaxEnt RL: *DrQ* [67], *CURL* [68], and a convolutional version of *SAC* [39].

Figure 3 (Bottom) shows the performance results after 1.5M environment steps. DMC tasks yield returns scaled within a set range, [0, 1000], which we use for normalization. Remarkably, also in this domain, SSPG attains state-of-the-art performance with statistically significant improvements over all baselines. Unlike for the Mujoco tasks, the other considered algorithms based on MaxEnt RL underperform as compared to the deterministic DrQv2, a result Yarats et al. [66] attributed to ineffective exploration. In contrast, SSPG yields performance gains *especially* on sparser reward tasks where the other baselines struggle (see App. D). These results validate the scalability of our framework to high-dimensional inputs and its ability to successfully complement MaxEnt RL.

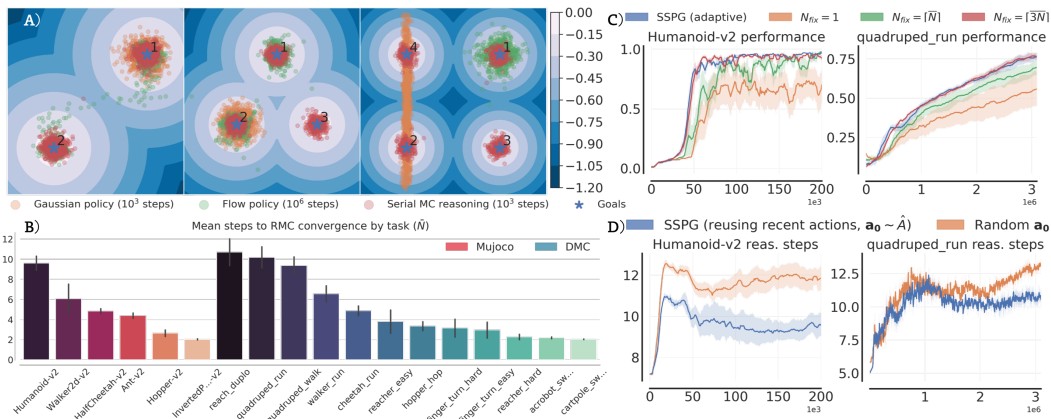

Figure 4: (**A**) Samples visualizations for learned policies in *positional bandits* of increasing complexity. (**B**) Mean number of *reasoning steps* required to reach convergence in each task with SSPG. (**C**) Mean performance ablating the adaptive strategy for detecting *reasoning convergence* and using fixed numbers of steps. (**D**) Number of *reasoning steps* throughout training with and without reusing recent actions as initial *action-beliefs*.

## 4.2 Properties of serial Markov chain reasoning

We test if theorized benefits of our framework (Sec. 3.5) hold in practical settings with deep networks and stochastic optimization. We provide further ablation studies and analysis of SSPG in App. E.

**1. Policy expressiveness.** First, we test the expressiveness of the behavior learned with SSPG using a Gaussian BT-policy. We design a series of single-step toy RL problems where the agent needs to position itself on a small 2D environment with a reward function based on unknown goal locations, which we name *positional bandits* (see App. C for details). The objective of these experiments is to isolate how our framework compares with traditional policies for MaxEnt RL to explore the environments and learn to match the true canonical distributions of returns. As displayed in Fig. 4 A, even in highly multi-modal positional bandits, the SS-policy successfully learns to visit all relevant goals with similar frequencies. Furthermore, quantizing the state space around the goals reveals that the relative RMC intuitively learns to *transition between action-beliefs that visit the different goals* as reasoning progresses, with a transition matrix matching a *cyclic permutation* (App. F). In comparison, a squashed Gaussian policy expectedly fails to capture the complexity of the canonical distribution, with samples either collapsing to a single mode or covering large suboptimal parts of the action space. We also show results for a policy based on normalizing flows [69, 70], modeled with a deep expressive network (App. C). After several attempts, we find these models require orders of magnitude more training iterations and data to learn any behavior that is more complex than a uni-modal distribution. Yet, even after increasing training by a factor of 1000, we still observe the flow policy distribution collapsing in the more complex positional bandits. We attribute our findings to training inefficiencies from a lack of proper inductive biases for flow models in the *non i.i.d.* RL problem setting [71]. In particular, as flows can assign arbitrarily low probability mass to some regions of the action space, initial local optima can greatly hinder future exploration, exacerbating coverage of the data buffer distribution in a vicious circle.

**2. Policy adaptivity.** Second, we examine the adaptivity of our framework for tackling decision-making problems with different complexities. We compare the average number of reasoning steps ($\bar{N}$) performed by SSPG for each task from Sec. 4.1 (Fig. 4 B). We identify a general correlation between task difficulty and reasoning computation, with complex robotic manipulation and humanoid locomotion problems requiring the most steps. By concentrating on two representative tasks, we validate the effectiveness of the reasoning process and our adaptive convergence detection strategy with an ablation study where we train SSPG using a *fixed* number of reasoning steps $N_{fix} \in \{1, \lceil \bar{N} \rceil, \lceil 3\bar{N} \rceil\}$. For the case $N_{fix} = 1$, which closely resembles traditional RL, we use double the UTD ratio to improve performance and offset any training-time gains from multi-step reasoning. As shown in Fig. 4 C, increasing $N_{fix}$ yields clear performance improvements, validating that agents can greatly benefit from performing longer reasoning processes. Furthermore, our adaptive SSPG

attains the same performance as $N_{fix} = \lceil 3\bar{N} \rceil$ and visibly outperforms $N_{fix} = \lceil \bar{N} \rceil$. These results show how different action selection problems require different amounts of *reasoning computation* and validate the practical effectiveness of our adaptive strategy to detect steady-state convergence. We obtain analogous findings for additional tasks and values of $N_{fix}$ in App. F.

**3. Solution reuse.** Last, we examine the effects of the *short-term action memory* buffer ($\hat{A}$) to sample initial action beliefs ($\mathbf{a_0}$) in two tasks. We evaluate ablating $\hat{A}$, randomly re-initializing $\mathbf{a_0}$ from a uniform distribution. While there are only minor differences performance-wise between the two approaches (App. F), sampling $\mathbf{a_0}$ from the short-term action memory considerably decreases the number of reasoning steps for convergence (Fig. 4D). Moreover, we observe the gap in reasoning efficiency expands throughout training as the agent's steady-state behavior further improves for the target task. This result validates that a simple temporal heuristic can provide considerable efficiency benefits, amortizing the additional computational cost of our powerful new framework.

# 5    Related work

There have been several prior attempts to extend ubiquitous Gaussian policies [13, 39, 72, 73] with simple normalizing flows [69, 70], both to improve expressiveness [74, 75] and to instantiate behavior hierarchies [76]. Yet, the expressiveness of normalizing flows is coupled with some training challenges [71], which we show can lead to premature convergence to suboptimal solutions in RL (Sec. 4.2). Other works also considered entirely replacing policy models with gradient-free [4] or gradient-based optimization over the predicted values [77]. Marino et al. [62] similarly considered *learning* an optimizer to *infer* Gaussian behavior [28] with iterative amortization [63]. However, while all these works consider alternative modeling of agent behavior, they are still based on the traditional RL framework of representing decision-making as the output of a *fixed* process. Instead, our work entails a conceptually different approach and enables implicit modeling of agent behavior as the result of an *adaptive* reasoning process, orthogonally providing agents also with additional flexibility to scale computation based on the properties of each individual input state.

Outside RL, there have been efforts to model generation processes with parameterized Markov chains learned to revert fixed noise injection processes acting on data [78–82]. Based on this framework, diffusion models [83–85] recently achieved remarkable results for image generation [85, 86]. While applied to inherently different problem settings, these works share some conceptual resemblances with our framework and highlight the vast *scaling* potential of implicit modeling.

# 6    Conclusion

We introduced *serial Markov chain reasoning*, a novel framework for modeling agent behavior in RL with several benefits. We showed our framework allows an agent to 1) learn arbitrary continuous action distributions, 2) flexibly scale computation based on the complexity of individual action-selection decisions, and 3) re-use prior solutions to accelerate future reasoning. Hence, we derived *SSPG* an off-policy maximum entropy RL algorithm for serial Markov chain reasoning, achieving state-of-the-art performance on two separate continuous control benchmarks. While for problems with discrete action spaces simple multinomial policy distributions already provide unlimited expressivity, we note that the inherent computational adaptivity of our framework could still yield benefits over traditional fixed policies in these settings. Furthermore, we believe our motivation and early results provide a strong argument for the future potential of serial Markov chain reasoning, even beyond off-policy RL and simulation tasks. We provide our implementation for transparency and to facilitate future extensions at `sites.google.com/view/serial-mcr/`.

# Acknowledgments

We thank Johannes Lutzeyer for providing valuable feedback on an earlier draft of this work. Edoardo Cetin would like to acknowledge the support from the Engineering and Physical Sciences Research Council [EP/R513064/1]. Oya Celiktutan would also like to acknowledge the support from the LISI Project, funded by the Engineering and Physical Sciences Research Council [EP/V010875/1]. Furthermore, we thank Toyota Motor Europe and Toyota Motor Corporation for providing support towards funding the utilized computational resources.

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
