# OpenReview forum: "Policy Gradient With Serial Markov Chain Reasoning"
_NeurIPS.cc/2022/Conference — NeurIPS 2022 Accept_

### Official Review · Reviewer_8GHv · 2022-07-08

**Rating:** 8
**Confidence:** 4
**Soundness:** 3 good
**Presentation:** 3 good
**Contribution:** 3 good

**Summary:**

The paper proposes a family of implicitly defined stochastic policies that arise as the steady-state distribution of a Markov Chain whose transition function is a neural net which takes an action, a system state and a noise vector and produces a new action. The authors motivate this family of policies in three ways: a) they can approach any distribution over the action space b) the computation required to plan an action can vary depending on the state c) planning can be shortened by initializing the Markov chain with recently used actions, which can be seen as a form of solution re-use. The method improves over all considered baselines in typical benchmarks.

**Questions:**

### Questions

* In eq. 13, the gradient term contains a product of gradients over the reasoning steps. Do the usual concerns about vanishing/exploding gradients apply here? Did you observe instability in practice?

* What is the real-time capability of these policies (in terms of deployment after training)?

**Limitations:**

These points are addressed well, aside from my comments above about computation time at deployment.

**Strengths And Weaknesses:**

### Strengths

* The idea reminds me of things like deep equilibrium models, but I believe the specific form here might have an advantage over those (stochasticity). I certainly haven't seen this type of model applied to RL, and I found it quite interesting.
* The paper has a good flow. Ideas, derivations and results are presented in a way that can be followed easily. Steps that would be complicated to parse at first reading are relegated to the appendix, which is good.
* The experimental results are both thorough and equivocally in favor of the model.
* The paper presents ablations that support the motivation for using a policy that refines itself over multiple steps (beyond that of improved performance). The experiments on simple positional bandits show that the model has an easier time accommodating multi-model distributions compared to normalizing flows, which would be the go-to model for something like this (aside from mixture density networks perhaps). Figure 4 b) and d) support the intuitions about multi-step reasoning. In OpenAI-Gym, Humanoid requires the most reasoning steps, while pendulum requires the least for instance. Re-using recent actions seems to bring an overall reduction in the number of reasoning steps, which is another thing that checks out.

### Weaknesses

* I believe the core weakness of the paper is the increase in computation. The authors analyze this point in terms of training time. I believe it should also be analyzed in terms of the cost of acting during deployment. How many times per second can we query such a policy? This remains somewhat unaddressed. I think it would also be interesting to compare SSPG to the other baselines in terms of the number of reasoning steps that are used (at the end of training, if we allow SSPG to be trained the usual way). At how many steps does SSPG outperform the others? There is a comparison of SSPG to itself in this fashion in the two hardest environments, but that doesn't exactly cover the same area.
* The increase in complexity over a standard policy is another weakness. SSPG adds some hyper-parameters and non-trivial implementation cost into the mix. There are also concerns about the assumptions that we have to make, such as the accuracy of the convergence criteria.

---

> ### Author Response · Authors · 2022-08-02
> **Responses to 8GHv 1/3**
>
> **Weaknesses**
>
> > 1)  believe the core weakness of the paper is the increase in computation. The authors analyze this point in terms of training time. I believe it should also be analyzed in terms of the cost of acting during deployment. How many times per second can we query such a policy? This remains somewhat unaddressed. I think it would also be interesting to compare SSPG to the other baselines in terms of the number of reasoning steps that are used (at the end of training, if we allow SSPG to be trained the usual way). At how many steps does SSPG outperform the others?
>
> (Also addressing **Question 2**: What is the real-time capability of these policies (in terms of deployment after training)?)
>
>
> Following the reviewer's suggestions, **we added Appendix D.4, which includes two new results**:
>
> First, **we reported the performance of SSPG when 'clipping' the maximum number of reasoning steps allowed for each action-selection only during evaluation**. As suggested, we evaluated the final checkpoints of agents learned by our unmodified SSPG, without any clipping. We consider four environments, Humanoid-v2, Ant-v2, cheetah\_run, and quadruped\_run for which SSPG displays different average reasoning requirements.
>
>
> | **Task/Algorithm** |   **SSPG**   | **SSPG, 1 reasoning step** | **SSPG, 4 max. reasoning steps** | **SSPG, 8 max. reasoning steps** |    **REDQ**   |
> |:------------------:|:--------:|:----------------------:|:----------------------------:|:----------------------------:|:---------:|
> |       Ant-v2       | 5513±238 |        5356±261        |           5501±114           |           5527±219           | 3792±1064 |
> |     Humanoid-v2    |  5148±51 |         4971±98        |            5120±56           |            5139±76           |  4721±648 |
> | **Task/Algorithm** |   **SSPG**   | **SSPG, 1 reasoning step** | **SSPG, 4 max. reasoning steps** | **SSPG, 8 max. reasoning steps** |   **DrQv2**   |
> |     cheetah_run    |  888±10  |         880±13         |            889±10            |             891±5            |   873±55  |
> |    quadruped_run   |  760±64  |         715±85         |            734±81            |            752±66            |  494±288  |
>
>
>
> As shown in the above table, clipping to as low as four reasoning steps only marginally affects SSPG performance, which still always surpasses the scores achieved by standard reinforcement learning baselines. Furthermore, SSPG is less affected by this form of deployment-only clipping than by fixing the number of reasoning steps for both training and evaluation phases (see Figure 4 C and Figure 12). We motivate this finding by noting that better capturing the canonical distribution of returns from the critic by performing additional reasoning steps has also very significant benefits during the training phase of off-policy algorithms. In particular, this allows the agent to achieve better exploration and more easily correct the critic in the areas of the action space where its predictions are erroneously optimistic. When training with the unmodified SSPG algorithm, this benefit is still fully retained, justifying the superior performance compared to our previous fixed-steps modification.

---

> > ### Author Response · Authors · 2022-08-02
> > **Responses to 8GHv 2/3**
> >
> > **Weaknesses**
> >
> > > 1) *(Continued)*
> >
> > Second, together with the average training times in Appendix D.3, **we now also report and discuss the average rollout times during deployment of each of our implementations**:
> >
> >
> > | **OpenAI Gym Mujoco**              | **Deployment time (seconds) for 1000 env. steps** |
> > |--------------------------------|:---------------------------------------------:|
> > | Random (only simulation)       |                     0.583                     |
> > | SSPG                           |                     3.523                     |
> > | SSPG, 1 reasoning step         |                     2.462                     |
> > | SSPG, 4 max. reasoning steps   |                     2.917                     |
> > | SSPG, 8 max. reasoning steps   |                     3.257                     |
> > | SAC-20                         |                     2.451                     |
> > | IAPO (Original implementation) |                     5.657                     |
> > | REDQ (Original implementation) |                     2.671                     |
> > | **DeepMind Control (pixels)**      | **Deployment time (seconds) for 1000 env. steps** |
> > | Random (only simulation)       |                     0.997                     |
> > | SSPG                           |                     1.701                     |
> > | SSPG, 1 reasoning step         |                     1.637                     |
> > | SSPG, 4 max. reasoning steps   |                     1.693                     |
> > | SSPG, 8 max. reasoning steps   |                     1.699                     |
> > | DrQv2                          |                     1.604                     |
> >
> > As shown in the above table, using SSPG does increase the average rollout time over standard reinforcement learning baselines. However, the additional time required for action-selection scales sub-linearly with the number of reasoning steps, and appears to be dominated by other fixed costs, such as simulating the environment and converting observations to tensor objects. This is in contrast with the other, more expensive, iterative baseline, IAPO [1], which performs gradient-based optimization at each acting step. Differences with standard reinforcement learning are even more marginal in the visual DeepMind Control environments, where the most expensive part of the computation is from encoding the observation with a convolutional encoder (which needs to occur only once before performing the reasoning steps). Moreover, we note that in many real-world applications, the additional acting cost would still be greatly inferior to the actuation time costs when using distributed hardware. However, clipping the number of reasoning steps still remains a valuable option, as examined in the previous experiment.
> >
> > > 2) The increase in complexity over a standard policy is another weakness. SSPG adds some hyper-parameters and non-trivial implementation cost into the mix. There are also concerns about the assumptions that we have to make, such as the accuracy of the convergence criteria.
> >
> > Since serial Markov chain reasoning is a novel framework, where action-selection is an adaptive iterative process, it adds additional implementation complexity compared to traditional reinforcement learning. Yet, as listed in Tables 1 and 2 and described in Appendix C, our framework actually introduces only four additional hyper-parameters to popular algorithms, which did not require any tuning and were kept unchanged in both benchmarks used for evaluation. Furthermore, in Appendix E we show that the performance of SSPG is quite insensitive to most reasonable choices for two of these hyper-parameters. Following the reviewer's comment, **we now explicitly address the additional implementation complexity and better remark SSPG's robustness with respect to the new hyper-parameters in Appendix C**.

---

> > > ### Author Response · Authors · 2022-08-02
> > > **Responses to 8GHv 3/3**
> > >
> > > **Questions**
> > >
> > >  > 1) In eq. 13, the gradient term contains a product of gradients over the reasoning steps. Do the usual concerns about vanishing/exploding gradients apply here? Did you observe instability in practice?
> > >
> > > As noted by the reviewer, our new policy gradient computation involves a sum over $N+1$ gradient terms (Equation 13), indexed by $n=0,1,\dots,N$. Each of these terms represents the expected local effect that the BT-policy's action has $n$ reasoning steps into the reasoning process and involves a product of $n$ gradients over the RMC's dynamics. In Appendix A.2, we prove that this sum exponentially converges, which means that for its further terms, as $n$ increases, we always get a 'vanishing' gradient effect due to the noted product (Equation 23). Intuitively, this is caused by 'reaching' the steady-state distribution which is independent of the selected starting reasoning action that initialized the chain. However, in our case, this vanishing effect does not yield instabilities and is actually desired, since it allows us to approximate tractably the original infinite sum in Equation 9 by ignoring later terms. Following the reviewer's question, **we extended Section 3.2 of the camera-ready version to better express these considerations, with references to the vanishing gradients phenomenon**.
> > >
> > >  > 2) What is the real-time capability of these policies (in terms of deployment after training)?
> > >
> > > Please, see our response to **Weakness 1**.
> > >
> > > **References**
> > >
> > > [1] Marino, Joseph, et al. "Iterative amortized policy optimization." Advances in Neural Information Processing Systems 34 (2021): 15667-15681.

---

> > > > ### Comment · Reviewer_8GHv · 2022-08-04
> > > > **Thank you for your response**
> > > >
> > > > Thank you for the detailed response and the additional experiments.

---

> > > > > ### Author Response · Authors · 2022-08-05
> > > > > **Further response to 8GHv**
> > > > >
> > > > > We would like to very much thank reviewer 8GHv for raising targeted feedback and suggesting relevant new experiments for improving our work.

---

### Official Review · Reviewer_Mb6P · 2022-07-11

**Rating:** 7
**Confidence:** 4
**Soundness:** 3 good
**Presentation:** 4 excellent
**Contribution:** 4 excellent

**Summary:**

This paper proposes a novel framework for representing decision making in an RL problem as an iterative reasoning process based on serial Markov chains. The authors derive a policy gradient theorem for their framework in the standard RL setting, as well as the MaxEnt RL setting. The empirical evaluation shows improved performance in both state and pixel observations and demonstrates the expressivity of the proposed policy representation.


**Questions:**

My major question is regarding the point discussed in the weaknesses. I have noted it here for easier reference. The rest of the questions are mostly minor comments/suggestions.

1. Can you clarify equation (8) and explain why $\nabla_\theta Q_n^s(s, a) = 0$. I am not convinced by the argument of the local approximation, as detailed above.
2. I suggest the authors give titles to the theorems for better clarification and better guidance of the reader. An inexperienced reader may be lost within the heavy math.
3. The parameterization of the policy is sometimes written (denoted as $\pi_\theta^b$) and sometimes dropped (denoted as $\pi^b$ or $\pi^s$). This can confuse and mislead the reader.
4. This is for my curiosity and I was not expecting to find this answer in the paper. Can recurrent neural networks be used to parameterize the RMC? I believe prior work has made the connection between RNNs and Markov chain reasoning.
5. Figure 1 is a very generic figure without giving much information about the method. I suggest the authors reconsider that and replace it with something more specific to their work.
6. The colors and font size on Figure 2 makes the text hard to read.


**Limitations:**

Yes, the authors have addressed the limitations and potential negative societal impact of their work.

**Strengths And Weaknesses:**

This is a solid paper with a clear and interesting idea, novel contribution, and a series of thoroughly conducted evaluations. I, however, have certain concerns regarding one of the key equations / definitions used by the authors which is the basis for all of the main derivations.

## Strengths

### 1. Novelty
The proposed method for obtaining policy gradients using serial Markov chain reasoning is novel, to the best of my knowledge. Furthermore, I believe the contributions are significant and relevant as the proposed framework is a core development of RL algorithms and can be potentially impactful.

### 2. Empirical Soundness
The empirical evaluation is thorough and sound. In particular, I appreciate using Rliable metrics and comparison against several strong baselines for both state and pixel observations. Finally, several ablation studies are conducted in Appendix E to shed more light onto the method and its hyperparameters.

### 3. Demonstration of useful properties of the proposed method
I greatly appreciate section 4.2 and correspondingly Appendix F. I have found the empirical results on policy expressiveness and adaptivity to be very convincing and a nice divergence from the standard comparison in the (deep) RL community that is usually concerned only with pure performance. Nevertheless, I do appreciate that the proposed method is outperforming SOTA algorithms.

## Weaknesses

### 1. Theoretical soundness
I have carefully checked the proofs of Lemma 3.1 and Theorem 3.2 and I acknowledge that the math is correct and sound (based on my understanding) **if** one agrees with the fact that $Q_n^s(s, a)$ is not a function of policy parameters $\theta$. However, I am not convinced this is the case. The concerning equation is equation (8) in which the authors define:
$$
Q_n^s(s, a) = \int_A \pi_n^b(a’|a,s) Q_\phi^s(s, a’)da’,     \text{   with    } \nabla_\theta Q_n^s(s, a) = 0
$$
First, I am confused with the **"with"** statement; is that an assumption, condition, or a result of the definition of $Q_n^s(s, a)$? Second, in L416 of Appendix, the authors write “[since] $Q_n^s(s, a)$ is a local approximation of the RMC with no dependence from $\theta$...” and continue the proof without again stating why that is the case.

My understanding is that the $\pi_\theta^b$ is paremetrized by $\theta$, hence $\pi_n^b$ is also a function of $\theta$ based on equation (4). In that case, I am interested to know why differentiating equation (8) does not yield the following result which is clearly non-zero:

$$ \nabla_\theta Q_n^s(s, a) = \int_A \nabla_\theta \pi_n^b(a' | a, s) Q_\phi^s(s, a') da' +  \pi_n^b(a' | a, s) \nabla_\theta  Q_\phi^s(s, a') $$

Given that equation (8) is the key for all the consequent derivations, including the main theoretical result of the paper, I am concerned with the theoretical soundness of the paper.

Note: With that said, I am hopeful that this is just a misunderstanding from my side and I am looking forward to hearing the authors’ response to elaborate and clear the confusion. Unfortunately for this reason, my rating is relatively low depite my acknowledgment of the strengths of the work. But I am eagerly looking forward to increasing it if the authors can explain the reasoning behind $\nabla_\theta Q_n^s(s, a) = 0$.

**Update after the rebuttal:** I acklowledge that I have been convinced by the argument of the authors and have increased my score from 6 (weak accept) to 7 (accept).

---

> ### Author Response · Authors · 2022-08-02
> **Responses to Mb6P 1/2**
>
> **Weaknesses**
>
> > 1) I have carefully checked the proofs of Lemma 3.1 and Theorem 3.2 and I acknowledge that the math is correct and sound (based on my understanding) if one agrees with the fact that $Q^s_n(s, a)$ is not a function of policy parameters $\theta$. However, I am not convinced this is the case...
>
> >(Also addressing **Question 1**: Can you clarify equation (8) and explain why $\nabla_\theta Q^s_n(s, a)=0$. I am not convinced by the argument of the local approximation, as detailed above.)
>
> Both parts of Equation 8 represent the definition of the members of the family of n-step extension to the Q-function. In particular, each term $Q_n^s$ can be viewed as a 'detached-gradient' version of $\int_A \pi_n^b(a'|a,s) Q^s_\phi(s,a') da'$. The meaning of this definition is that while the value of $Q_n^s$ equals the value of $\int_A \pi_n^b(a'|a,s) Q^s_\phi(s,a') da'$, its gradient with respect to $\theta$ does not. We represent this 'gradient independence' concept using the statement "with $\nabla_\theta Q^s_n(s, a) = \mathbf{0}$", *which is itself part of our definition* of $Q_n^s$, *and not a property of* $\int_A \pi_n^b(a'|a,s) Q^s_\phi(s,a') da'$. A way to interpret each $Q_n^s$ term is that its value is dependent on a 'local copy' of the RMC before each optimization step, whose value stays fixed with changes in the BT-policy (unlike $\int_A \pi_n^b(a'|a,s) Q^s_\phi(s,a') da'$). This property is also what we referred to when we stated in L416 that "$Q_n^s$ is a local approximation of the RMC that does not depend on $\theta$ (as defined in Eq. 8)".
>
> The reason we define the $Q_n^s$ terms this way is that, to prove Theorem 3.2, we have to recursively apply the product rule, resulting in relationships such as:
>
> $$\nabla_\theta E_{a\sim \pi^s_\theta(\cdot|s)}\left[Q^s(s, a)\right] = \nabla_\theta \int_A \int_A \pi^s_\theta(a|s) \pi^b_\theta(a'| a, s) Q^s(s, a') da' da$$
> $$= \int_A \pi^s_\theta(a|s) \nabla_\theta \int_A \pi^b_\theta(a'| a, s) Q^s(s, a') da' da
>         + \int_A (\nabla_\theta\pi^s_\theta(a|s)) \int_A \pi^b_\theta(a'| a, s) Q^s(s, a')$$
> $$=  E_{a\sim \pi^s_\theta(\cdot| s)} \left[ \nabla_\theta E_{a'\sim \pi^b_\theta(\cdot| a, s)}\left[Q^s(s, a')\right]\right]
>         + \int_A (\nabla_\theta\pi^s_\theta(a|s)) \int_A \pi^b_\theta(a'| a, s) Q^s(s, a') da' da. \quad (1) + (2)$$
>
> (This example is from below line 418 in Appendix A, following the first step in the recursion).
>
> Hence, our definition of the n-step extensions $Q_n^s$, allows us to write term (2) as an expectation, $\int_A (\nabla_\theta\pi^s_\theta(a|s)) \int_A \pi^b_\theta(a'| a, s) Q^s(s, a') da' da = \nabla_\theta E_{a\sim \pi^s_\theta(\cdot|s)}\left[Q_1^s(s, a)\right]$ (since $\nabla_\theta Q^s_n(s,a)=\mathbf{0}$). Writing term (2) as an expectation, allows us to simplify notation and more explicitly show how the relationship between $\nabla_\theta E_{a\sim \pi^s_\theta(\cdot|s)}\left[Q^s(s, a)\right]$ and $\nabla_\theta E_{a\sim \pi^s_\theta(\cdot|s)}\left[Q_1^s(s, a)\right]$ can be extended to $\nabla_\theta E_{a\sim \pi^s_\theta(\cdot|s)}\left[Q_n^s(s, a)\right]$ and $\nabla_\theta E_{a\sim \pi^s_\theta(\cdot|s)}\left[Q_{n+1}^s(s, a)\right]$ in Equation 19.
>
> After the reviewer's request for clarification, we realized how our notation can have potentially ambiguous interpretations without a thorough explanation. Therefore, **we added a new paragraph below Equation 8 to precisely explain with examples the meaning of our definition in the camera-ready revision**, summarizing the clarification above. We hope reviewer Mb6P will not hesitate to let us know if there are other aspects of the definition and nature of the n-step extensions that are still unclear, such that we can address them in future responses and further improve the paper.

---

> > ### Author Response · Authors · 2022-08-02
> > **Responses to Mb6P 2/2**
> >
> > **Questions**
> >
> > > 1) Can you clarify equation (8) and explain why $\nabla_\theta Q^s_n(s, a)=0$. I am not convinced by the argument of the local approximation, as detailed above.
> >
> > Please, see our response to **Weakness 1**.
> >
> > > 2) I suggest the authors give titles to the theorems for better clarification and better guidance of the reader. An inexperienced reader may be lost within the heavy math.
> >
> > Following the reviewer's suggestion, **we titled Lemma 3.1 as the 'steady-state convergence lemma' and Theorem 3.2 as the 'steady-state policy gradient Theorem.' When referring to these results, we now use both their new titles and their relative numbers to facilitate the reader's navigation** within the paper.
> >
> >  > 3) The parameterization of the policy is sometimes written (denoted as $\pi^b_\theta$) and sometimes dropped (denoted as $\pi^b$ or $\pi^s$). This can confuse and mislead the reader.
> >
> >
> > Following the reviewer's suggestion, **we replaced all occurrences of $\pi^b$/$\pi^s$ with $\pi^b_\theta$/$\pi^s_\theta$ after line 120** (when introducing the parameterized BT-policy $\pi^b_\theta$ for the first time).
> >
> >  > 4) This is for my curiosity and I was not expecting to find this answer in the paper. Can recurrent neural networks be used to parameterize the RMC? I believe prior work has made the connection between RNNs and Markov chain reasoning.
> >
> > In line with the reviewer's intuition, early versions of our framework considered using some recurrent memory components as input to the BT-policy. However, to preserve the Markov property, this modification would require considering an extended state space for the RMC, consisting of both action-beliefs and memory hidden states. Preliminary results showed that such extended state space makes convergence require an increased amount of reasoning steps, and optimization of the BT-policy more unstable. Following the reviewer's interest, **we added a new brief discussion of this particular extension to Appendix C**.
> >
> >  > 5) Figure 1 is a very generic figure without giving much information about the method. I suggest the authors reconsider that and replace it with something more specific to their work.
> >
> > Following the reviewer's suggestion, **we replaced Figure 1** with a larger figure in the camera-ready revision, **now showing different paths that the reasoning process can take within the depicted RMC and an additional dependency between the transitions of the RMC and the environment state** (reflecting the dependency of the BT-policy).
> >
> >  > 6) The colors and font size on Figure 2 makes the text hard to read.
> >
> > Following the reviewer's comment, **we increased the font size in Figure 2. We also increased the thickness of the arrows and used darker colors for the middle plot**.

---

> > > ### Comment · Reviewer_Mb6P · 2022-08-05
> > > **Response to Authors Rebuttal**
> > >
> > > I thank the authors for their response and incorporating the changes I suggested. Although, it seems that the updated version is identical to the initial submission.
> > >
> > > I am convinced by the authors’ response to my main concern about the reason behind $\nabla_\theta Q_n^s(s, a)$, and as stated in my original review, I have increased my rating from 6 to 7.

---

> > > > ### Author Response · Authors · 2022-08-05
> > > > **Further response to Mb6P**
> > > >
> > > > We would like to very much thank reviewer Mb6P for their time and for raising constructive feedback and questions. Currently, we uploaded the revised Appendix in the supplementary materials, containing all new results and discussion. We have incorporated other main text changes, as described in our responses, to the camera-ready version of our work, using the extra-page concession (rebuttal revisions still feature a strict 9-page limit).

---

### Official Review · Reviewer_ZW89 · 2022-07-11

**Rating:** 6
**Confidence:** 4
**Soundness:** 4 excellent
**Presentation:** 2 fair
**Contribution:** 3 good

**Summary:**

This paper introduces a new serial markov chain reasoning for representing agent behavior. Instead of using a policy, the agent selects an action by maintaining an action-belief and a belief transition policy. The authors derive a steady state policy gradient algorithm for estimating the belief transition policy in a MaxEnt RL environment. In addition, this paper also provide a method to determine the number of reasoning steps dynamically. Finally, some empirical experiments are presented to verify the performance of the proposed method in real applications.

**Questions:**

I have several questions. The authors state that the proposed reasoning Markov chain is more general compared to the traditional policy. And in the first section, you mentioned that the agent’s behavior can approximate any arbitrary distribution with simple parameterized transition functions.  I don’t fully get this point from the theoretical analysis. Is it true that a parameterized belief transition function can approximate any policy? Can you elaborate your statement a bit by referring to the theorems in the paper?
And since you are using the stationary distribution of a markov chain to define the action, you assumed that the markov chain is ergodic and every action is reachable in every state. I suggest the authors explain the assumptions first before presenting the method and theorems.


**Limitations:**

Yes. The authors adequately addressed the limitations.

**Strengths And Weaknesses:**

Strengths: This method presented in this paper is novel. It has a potential to solve some RL problems in a more expressive way. The idea of learning the action belief instead of the policy makes sense to me. The proof of the theorems is rigorous and looks right to me. The experiments presented in this paper are sufficient and well-designed. The results look promising as the SSPG algorithm outperforms others with significant gains.

Weakness: This paper is a bit hard to follow since some background knowledge is not clearly presented. Before you introduce how to apply the techniques, it is better to have a comprehensive description for them.

---

> ### Author Response · Authors · 2022-08-02
> **Responses to ZW89**
>
> **Weaknesses**
>
> > 1) This paper is a bit hard to follow since some background knowledge is not clearly presented. Before you introduce how to apply the techniques, it is better to have a comprehensive description for them.
>
> Following the reviewer's comment, **we added a new Background subsection to the camera-ready revision, which provides information about the basic components of Markov chains, the concepts of steady-state convergence, and high-level background and intuitions regarding the Gelman-Rubin convergence diagnostic** (following [1, 2]). If reviewer ZW89's comment was also referring to any additional component of our methodology, we hope they will not hesitate to point them out, such that we can promptly address them in the paper.
>
>
> **Questions**
>
>  > 1) The authors state that the proposed reasoning Markov chain is more general compared to the traditional policy. And in the first section, you mentioned that the agent’s behavior can approximate any arbitrary distribution with simple parameterized transition functions. I don’t fully get this point from the theoretical analysis. Is it true that a parameterized belief transition function can approximate any policy? Can you elaborate your statement a bit by referring to the theorems in the paper?
>
> In Section 3.5, we explain that "the distribution of agent behavior given by the SS-policy, $\pi^s$, is a mixture model with potentially infinitely many components. Hence, even a simple Gaussian parameterization of the BT-policy $\pi^b$ (i.e., the parameterization of each such component) enables agent behavior to approximate any action distribution to arbitrary precision [3]" [215-218]. Following the reviewer's clarification request, **we extended our explanation by explicitly noting the property that a Gaussian mixture model is a universal approximator of densities**, pointing readers to an additional modern reference for this statement [4, page 65]. Furthermore, **in Appendix A, we added a new subsection that also references Theorem 3.2 from Nestoridis et al. [5], extending this property to arbitrary mixture models when approximating distributions defined on compact spaces**. We hope our extended discussion will better emphasize the rationale behind the unlimited representation power of our framework.
>
>
>  > 2) And since you are using the stationary distribution of a markov chain to define the action, you assumed that the markov chain is ergodic and every action is reachable in every state. I suggest the authors explain the assumptions first before presenting the method and theorems.
>
>
> Lemma 3.1 relies on the assumption that the action space is compact and that $\inf \{\pi^{b}(a'|a, s): a', a\in A\} > 0$, i.e., that the belief transition policy has some positive minimum density in the whole action-space (lines 393-394 in Appendix A). These assumptions are clearly fulfilled by the action spaces of the considered environments (being closed k-cells $[1, -1]^{|A|}$) and the Gaussian parameterization of our BT-policy. In Appendix A, we detail how these properties make the entire action space a *small set* [6, 7], This property implies that the Markov chain is ergodic and that every action-belief is reachable from every state, as noted by the reviewer.
>
> Following the reviewer's suggestions, **we now detail our assumptions directly when introducing Lemma 3.1 in the main text of the camera-ready ready revision**, to improve the clarity of our exposition. Furthermore, **we also extended the description of the implications of our assumptions in Appendix A, specifically mentioning the ergodicity, irreducibility, and aperiodicity properties** of the resulting reasoning Markov chain.
>
>
> **References**
>
> [1] Gelman, Andrew, and Donald B. Rubin. "Inference from iterative simulation using multiple sequences." Statistical science (1992): 457-472.
>
> [2] Brooks, Stephen P., and Andrew Gelman. "General methods for monitoring convergence of iterative simulations." Journal of computational and graphical statistics 7.4 (1998): 434-455.
>
> [3] Everitt, Brian. Finite mixture distributions. Springer Science & Business Media, 2013.
>
> [4] Goodfellow, Ian, Yoshua Bengio, and Aaron Courville. Deep learning. MIT press, 2016.
>
> [5] Nestoridis, Vassili, and Vangelis Stefanopoulos. Universal series and approximate identities. Technical Report TR-28-2007, Department of Mathematics and Statistics, University of Cyprus, 2007.
>
> [6] Orey, Steven. Limit theorems for Markov chain transition probabilities. London: van Nostrand, 1971.
>
> [7] Nummelin, Esa. General irreducible Markov chains and non-negative operators. No. 83. Cambridge University Press, 2004.

---

### Official Review · Reviewer_NxbF · 2022-07-12

**Rating:** 7
**Confidence:** 4
**Soundness:** 3 good
**Presentation:** 3 good
**Contribution:** 3 good

**Summary:**

This paper considers policies expressed as a reasoning Markov chain over actions in the current state that refines its choice of actions until arriving at a steady-state distribution over actions. In this framework the authors learn the number of steps to take before outputting an action and outline a policy gradient theorem that can be leveraged to differentiate through this process. The reason for learning this kind of policy seems to be in order to increase the expressiveness of distributions with multiple modes in the case of continuous action spaces. The authors present experiments demonstrating superior performance to baselines in 6 MuJoCo and 12 DMLab continuous control domains.

**Questions:**

- Can you compare what is achieved by your approach more directly to normalizing flows? The main advantage I understood from the related work section is allowing for a dynamic amount of computation. Am I missing something? Can there be an empirical comparison with normalizing flows?
- What are the “mild assumptions” used in the proof of Lemma 3.1? Even looking through the proof in the appendix, it is not 100% clear to me what these assumptions are. Based on my experience with the Markov chain literature I was expecting something related to stochasticity of transitions or irreducibility and aperiodicity of the Markov chain etc.
- Can you elaborate more about how information reuse is a benefit of this framework? It seems more like a necessity to make it practical? Is this meant in comparison with normalizing flows?

**Limitations:**

It seems that the authors mostly highlight value in their work with respect to continuous action space domains. Could this approach also provide value in domains with discrete action spaces?

**Strengths And Weaknesses:**

Strengths:
- The authors propose an adaptive iterative reasoning process with the goal of allowing for more complex multi-modal action space distributions for problems with continuous action spaces.
- The authors compare to a solid set of baselines across 6 mujoco domains and 12 DMLab domains while also providing nice qualitative experiments and relevant ablation experiments.
- The approach is a bit outside the box from recent research with the fact that it works well striking me as somewhat surprising on the surface.
- The derivation of Theorem 3.2 seems involved beyond the standard policy gradient derivation.

Weaknesses:
- The authors only provide an intuitive rationale for their proposed strategy, on the surface it seems a bit weird to iteratively update actions in this way. I understand that it makes the distribution more expressive, but it is not clear to me why this direction is preferable to normalizing flows or some adaptive version of normalizing flows.
- "Steady-state policy gradient" is a confusing term in light of the RL literature. I get what you mean upon reading, but it is confusing with the policy gradient for continuing environments which is defined over a different steady-state distribution (i.e. Sutton and Barto, 2018 Section 13.6).
- It would be nice to provide readers with more information regarding how convergence to the steady-state distribution was determined for those not familiar with pseudo scale reduction factors.

---

> ### Author Response · Authors · 2022-08-02
> **Responses to NxbF 1/3**
>
> **Weaknesses**
>
> > 1) The authors only provide an intuitive rationale for their proposed strategy, on the surface it seems a bit weird to iteratively update actions in this way. I understand that it makes the distribution more expressive, but it is not clear to me why this direction is preferable to normalizing flows or some adaptive version of normalizing flows.
>
> As also mentioned in reviewer NxbF's Question 1, one of the main advantages of our framework over traditional policies (either modeled by simple Gaussians or by normalizing flows) is its ability to adaptively scale with the difficulty of each decision-making problem. Furthermore, in the first part of Section 4.2, we compare the policy expressiveness recovered by different algorithms on our toy 'positional bandits' tasks. In this experiment, we compare our SSPG framework with traditional maximum entropy reinforcement learning algorithms making use of either a Gaussian policy or a powerful inverse autoregressive flow [1]. We find that only SSPG consistently recovers appropriate action distributions that reflect the true returns of the considered problems. While, in principle, also the normalizing flow policy can learn to represent complex distributions, we show that this baseline experiences representation collapse in the more complex positional bandits. This collapse is present even after training for orders of magnitudes more iterations and with wider policy networks. We attribute our findings to the optimization difficulties of training normalizing flows training with non-i.i.d data which characterizes the reinforcement learning settings [2]. In particular, we explain that initial local optima of the flow model can assign arbitrarily low probability mass to some regions of the action space and greatly hinder future exploration, exacerbating coverage of the data buffer distribution in a vicious circle. Following the reviewer's remark, **we extended Section 5 of the camera-ready revision with a summary of the conceptual and empirical advantages of our new framework over flow-based policies**, as detailed in this response.
>
> > 2) "Steady-state policy gradient" is a confusing term in light of the RL literature. I get what you mean upon reading, but it is confusing with the policy gradient for continuing environments which is defined over a different steady-state distribution (i.e. Sutton and Barto, 2018 Section 13.6).
>
> We thank the reviewer for pointing out this potential source of confusion. We now explicitly address this when first introducing our steady-state policy $\pi^s$ in Section 3.1 of the camera-ready revision. In particular, **we clarify that our use of 'steady-state distribution' is over the space of action-beliefs in the RMC** and is unrelated to the term's use in Sutton & Barto's seminal book.
>
>  > 3) It would be nice to provide readers with more information regarding how convergence to the steady-state distribution was determined for those not familiar with pseudo scale reduction factors.
>
> Following the reviewer's suggestion, **we added a new background subsection to the camera-ready revision, which provides additional preliminary information about the basic components of Markov chains, the concept of steady-state convergence, and high-level intuitions regarding the GR convergence diagnostic and the PSRF** (following [5, 6]). We believe this new Section will facilitate unfamiliar readers and complement our mathematical description of how convergence is computed in Section 3.4.

---

> > ### Author Response · Authors · 2022-08-02
> > **Responses to NxbF 2/3**
> >
> > **Questions**
> >
> >  > 1) Can you compare what is achieved by your approach more directly to normalizing flows? The main advantage I understood from the related work section is allowing for a dynamic amount of computation. Am I missing something? Can there be an empirical comparison with normalizing flows?
> >
> > Following the reviewer's suggestions, **we implemented additional baselines for our main performance evaluation section that extend REDQ and DrQv2 with the normalizing flow policy model considered in Section 4.2**. We summarize these results in the table below and Appendix D of the rebuttal revision (collected over 3 seeds). **We will collect the remaining seeds and add the Rliable analysis for the camera-ready version of our paper**. Our new results show that incorporating flows provides more limited and inconsistent performance benefits as compared to incorporating our serial Markov chain reasoning framework. This is in line with our motivation and with results reported in prior off-policy reinforcement learning work (e.g., [3, 4]) which show flows provide marginal gains over standard Gaussian or deterministic policies.
> >
> > | **Evaluation milestone** | 100.0K frames | 100.0K frames | 100.0K frames | 200.0K frames | 200.0K frames | 200.0K frames |
> > |--------------------------|:-------------:|:-------------:|:-------------:|:-------------:|:-------------:|:-------------:|
> > | **Task/Algorithm**       |      SSPG     |   REDQ-FLOW   |      REDQ     |      SSPG     |   REDQ-FLOW   |      REDQ     |
> > | Invertedpendulum-v2      |     1000±0    |     834±69    |     1000±0    |     1000±0    |     920±80    |     1000±0    |
> > | Hopper-v2                |    3314±68    |    2838±225   |    2994±510   |    3487±87    |    2829±593   |    3060±617   |
> > | Walker2d-v2              |    4428±230   |   2842±1168   |   1989±1003   |    4793±186   |   3905±1000   |    2969±861   |
> > | Halfcheetah-v2           |    8897±496   |    8395±981   |    5613±436   |   10309±653   |   9324±1096   |    6633±568   |
> > | Ant-v2                   |    5163±275   |   3743±1304   |   3132±1243   |    5513±238   |    4684±912   |   3792±1064   |
> > | Humanoid-v2              |    4992±140   |   3666±1681   |    1402±657   |    5148±51    |    4742±361   |    4721±648   |
> >
> >
> > | **Evaluation milestone** | 1.5M frames | 1.5M frames | 1.5M frames | 3.0M frames | 3.0M frames | 3.0M frames |
> > |--------------------------|:-----------:|:-----------:|:-----------:|:-----------:|:-----------:|:-----------:|
> > | **Task/Algorithm**       |     SSPG    |  DrQv2-FLOW |    DrQv2    |     SSPG    |  DrQv2-FLOW |    DrQv2    |
> > | Acrobot swingup          |    218±49   |    294±22   |    272±40   |    371±41   |    428±30   |    422±48   |
> > | Cartpole swingup sparse  |    797±43   |   548±388   |   478±391   |    837±15   |   561±397   |   503±411   |
> > | Cheetah run              |    755±47   |    781±34   |    781±32   |    888±10   |    832±68   |    873±55   |
> > | Finger turn easy         |   794±127   |   752±115   |   757±156   |    974±6    |    920±60   |    932±43   |
> > | Finger turn hard         |   637±138   |   503±159   |   506±229   |    945±42   |    841±24   |    913±60   |
> > | Hopper hop               |    246±28   |   156±114   |   200±102   |    344±28   |   194±144   |   239±123   |
> > | Quadruped run            |    570±22   |    606±62   |   402±213   |    760±64   |    738±23   |   494±288   |
> > | Quadruped walk           |    855±23   |    783±11   |   591±271   |    888±22   |    925±4    |    905±44   |
> > | Reach duplo              |    221±7    |    218±9    |    219±7    |    218±9    |    228±1    |    228±1    |
> > | Reacher easy             |    978±4    |    974±1    |    973±3    |    982±3    |    981±1    |    954±22   |
> > | Reacher hard             |    913±77   |    898±16   |   802±113   |    974±6    |    948±22   |    944±25   |
> > | Walker run               |    634±16   |   559±269   |   568±273   |    738±7    |   610±293   |   616±297   |
> > | **Average score**        |  **634.79** |    589.32   |    545.72   |    **743.32**   |    683.82   |    668.60   |
> > | **Median score**         |  **695.85** |    582.43   |    537.37   |    **862.73**   |    784.91   |    744.66   |

---

> > > ### Author Response · Authors · 2022-08-02
> > > **Responses to NxbF 3/3**
> > >
> > > **Questions**
> > >
> > >  > 2) What are the “mild assumptions” used in the proof of Lemma 3.1? Even looking through the proof in the appendix, it is not 100% clear to me what these assumptions are. Based on my experience with the Markov chain literature I was expecting something related to stochasticity of transitions or irreducibility and aperiodicity of the Markov chain etc.
> > >
> > > Lemma 3.1 relies on the assumption that the action space is compact and that $\inf \{\pi^{b}(a'|a, s): a', a\in A\} > 0$, i.e., that the belief transition policy has some positive minimum density in the whole action-space (lines 393-394 in Appendix A). These assumptions are clearly fulfilled by the action spaces of the considered environments (being closed k-cells $[1, -1]^{|A|}$) and the Gaussian parameterization of our BT-policy. In Appendix A, we detail how these properties make the entire action space a *small set* [7, 8], which implies both the irreducibility and aperiodicity of the Markov chain.
> > >
> > > To improve the clarity of our exposition, **we now detail our assumptions directly when introducing Lemma 3.1 in the main text of the camera-ready revision**. Furthermore, **we also extended the description of the implications of our assumptions in Appendix A, specifically mentioning the ergodicity, irreducibility, and aperiodicity properties** of the resulting reasoning Markov chain.
> > >
> > >  > 3) Can you elaborate more about how information reuse is a benefit of this framework? It seems more like a necessity to make it practical? Is this meant in comparison with normalizing flows?
> > >
> > > In Section 3.3, we describe how we can use recent action-beliefs stored in the *short-term action memory*, $\hat{A}$, to initialize the reasoning Markov chain. In Paragraph 3 of Section 4.2, we empirically demonstrate that using our short-term action memory buffer considerably reduces the number of reasoning steps for convergence. Hence, this practice enables us to make use of the temporal consistency properties of common MDPs to accelerate reasoning, retaining the increased performance of the serial Markov chain framework while reducing its computational overhead.
> > >
> > > Following the reviewer's request for clarifications, **we extended Section 4.2 of the camera-ready revision to explain that the benefits of information reuse are specific to amortizing the computational costs of our more powerful new framework**.
> > >
> > > **Limitations**
> > >
> > > > It seems that the authors mostly highlight value in their work with respect to continuous action space domains. Could this approach also provide value in domains with discrete action spaces?
> > >
> > > In environments with discrete action spaces, multinomial distributions can already tractably express arbitrary policy distribution. However, our framework could still provide efficiency and performance benefits over standard reinforcement learning methods from its ability to adaptively scale the number of reasoning steps with the difficulty of individual action selection problems. We believe such benefits are likely to be more relevant in environments with numerous possible actions, but leave such investigation to future work. Following the reviewer's comment, **we added a brief discussion of this limitation of our work in Section 6** for the camera-ready revision.
> > >
> > > **References**
> > >
> > > [1] Kingma, Durk P., et al. "Improved variational inference with inverse autoregressive flow." Advances in neural information processing systems 29 (2016).
> > >
> > > [2] Kobyzev, Ivan, Simon JD Prince, and Marcus A. Brubaker. "Normalizing flows: An introduction and review of current methods." IEEE transactions on pattern analysis and machine intelligence 43.11 (2020): 3964-3979.
> > >
> > > [3] Mazoure, Bogdan, et al. "Leveraging exploration in off-policy algorithms via normalizing flows." Conference on Robot Learning. PMLR, 2020.
> > >
> > > [4] Marino, Joseph, et al. "Iterative amortized policy optimization." Advances in Neural Information Processing Systems 34 (2021): 15667-15681.
> > >
> > > [5] Gelman, Andrew, and Donald B. Rubin. "Inference from iterative simulation using multiple sequences." Statistical science (1992): 457-472.
> > >
> > > [6] Brooks, Stephen P., and Andrew Gelman. "General methods for monitoring convergence of iterative simulations." Journal of computational and graphical statistics 7.4 (1998): 434-455.
> > >
> > > [7] Orey, Steven. Limit theorems for Markov chain transition probabilities. London: van Nostrand, 1971.
> > >
> > > [8] Nummelin, Esa. General irreducible Markov chains and non-negative operators. No. 83. Cambridge University Press, 2004.

---

> > > > ### Comment · Reviewer_NxbF · 2022-08-09
> > > > **Re: Responses to NxbF**
> > > >
> > > > Thank you for your very thorough response to my key points of confusion. I feel that many of my main concerns have now been addressed and I have raised my score accordingly.

---

> > > > > ### Author Response · Authors · 2022-08-09
> > > > > **Further response to NxbF**
> > > > >
> > > > > We would like to very much thank reviewer NxbF for providing constructive criticism and suggesting relevant new comparisons.

---

### Author Response · Authors · 2022-08-02
**Responses overview and paper updates**

We are glad that all the reviewers appreciated the novelty, soundness, and empirical value of our work. Moreover, we are very grateful for the specific and impactful feedback provided. We have posted detailed replies to the reviews that specifically address each question and concern, where we **emphasized** the resulting changes for the camera-ready version of our work. We summarize some of the most relevant such changes below:

- We implemented extensions to REDQ and DrQv2, using powerful normalizing flow policies. We used these extensions as additional baselines for the performance evaluation of SSPG in the OpenAI Gym and DeepMind Control benchmarks.

- We detailed the assumptions of Lemma 3.1 in the main text and provided an extended description of their implications.

- We extended Section 2 with some preliminary concepts of Markov chains and high-level background and intuitions regarding the Gelman-Rubin convergence diagnostic.

- We added a new paragraph below Equation 8 to precisely explain, with examples, the meaning of our definition of the family of n-step extensions to the Q-function.

- We added results for an additional modification of SSPG, 'clipping' the maximum number of reasoning steps allowed for each action selection to fixed values after standard training.

- We added results and a discussion for the average rollout time during deployment of each of our implementations.

The camera-ready version of our work features many of these changes in the main text, given the extra page concession. While all our responses are self-contained and describe all such changes, we also added key new results and additional discussion to our updated Appendix for the rebuttal revision. We remain open to preparing and uploading an additional shorter 9-page preliminary version of our work before the camera-ready deadline, if required.

---

### Meta-Review · Area_Chair_GrED · 2022-08-24

**Recommendation:** Accept
**Confidence:** Certain

**Metareview:**

The paper makes a solid algorithmic contribution to RL that is extensively evaluated empirically and is a certain accept.

**Award:**

No

---

### Decision · Program_Chairs · 2022-09-14

Accept